# MAMBAMATCH: LEARNING TWO-VIEW CORRESPONDENCES WITH SELECTIVE STATE SPACES

## ABSTRACT

Two-view correspondence learning aims to discern true and false correspondences between image pairs by recognizing their underlying different information. Previous methods either treat the information equally or fail to discard the superfluous information of false correspondences, tending to be invalid in practical scenarios. Therefore, inspired by Mamba's inherent competence of selectivity, we propose MambaMatch as a Mamba-based correspondence filter to selectively mine information from true correspondences and to dispose of the potentially interfering information of false correspondences. Specifically, the selection is achieved by adaptively adjusting model parameters in a high-dimensional latent space, which also avoids attention leakage and implements context compression, ensuring the precise and efficient exploitation of pertinent information. Meanwhile, channel awareness is tailored to serve as a complementary aspect of comprehensive information acquisition. Moreover, we design a novel local-context enhancement module to capture reasonable local context that is crucial for correspondence pruning. Extensive experiments demonstrate that our approach outperforms existing state-of-the-art methods on several visual tasks while saving time and space costs.

## 1 INTRODUCTION

Two-view correspondence learning that finds sparse matches and estimates geometric relationships for image pairs is a fundamental and crucial task in computer vision. It is of paramount importance for many downstream tasks such as image retrieval (Tolias et al., 2016), Structure from Motion (Saputra et al., 2018), and simultaneous localization and mapping (Mur-Artal et al., 2015). The typical pipeline is divided into two stages, namely generating a putative correspondence set and removing false matches (*i.e.* outliers) (Ma et al., 2021). However, constrained by the limited discriminative ability of descriptors, the sparse and irregular matches in the putative set usually contain a large number of outliers, and re-

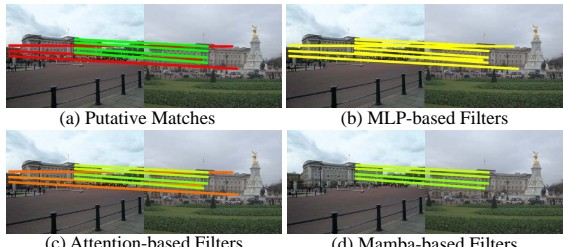

(a) Putative Matches      (b) MLP-based Filters

(c) Attention-based Filters      (d) Mamba-based Filters

Figure 1: Illustration of different methods for handling putative matches. False matches are shown in red (—), and correct ones in green (—). The transition of the yellow hue from closer to red to closer to green signifies a progression from lower to higher weights, meaning the model is more likely to consider the match as an inlier.

moving them is imperative for image matching. Therefore, in this paper, we focus on outlier rejection to maintain true correspondences (*i.e.* inliers), which facilitates accurately estimating the true model.

In prior research, nascent outlier rejection methods such as RANSAC (Fischler & Bolles, 1981), MAGSAC++ (Barath et al., 2020) and VFC (Ma et al., 2014) have demonstrated good performance in situations where outliers are relatively minor. However, they often fail in real-world scenarios with numerous outliers. Among the burgeoning learning-based approaches, Yi et al. (2018) and some subsequent works (Zhang et al., 2019; Zhao et al., 2021; Liu et al., 2021) regard the corresponding pruning as a binary classification problem. Yet, as shown in Figure 1(b), they employ multi-layer perceptrons (MLPs) that treat each input equally, which fails to adequately discern the differences in information resulting from inliers and outliers. Sun et al. (2020) customizes to give different

attention to diverse inputs through explicit weights, but this simplistic approach is insufficient for handling complex scenarios. Additionally, some vanilla attention-based methods (Liu & Yang, 2023; Li et al., 2024) modulate the importance of information by constructing feature maps for the inputs and assigning different weights. However, this soft attention is allocated across all data throughout, leading to attention leakage and the retention of redundant context (*i.e.* incapacity to compress context) as presented in Figure 1(c), which degrades the performance. Furthermore, attention-based methods are impeded by quadratic complexity, resulting in inefficiency during both inference and training. Therefore, we consider the following questions: (i) *How to selectively treat inliers and outliers, and allocate attention appropriately to different points?* (ii) *Can we fully capture the information and compress the context into a more compact representation as much as possible?*

Recently, a novel architecture termed Selective State Space Model, *i.e.* Mamba (Gu & Dao, 2023), has gained widespread adoption in visual tasks (Liu et al., 2024b; Zhu et al., 2024). One of its key advantages is the selectivity, enabling models to focus on or disregard specific inputs. This may offer a pathway to address (i). Furthermore, it achieves optimization of complexity through compressing context into a compact representation and employs hardware scanning methods. This presents a potential avenue for addressing (ii). Inspired by this, we seek to introduce Mamba to tackle the problem of two-view correspondence learning. However, most existing Mamba-based visual models are tailored for tasks involving regular images as input (Guo et al., 2024; Zhu et al., 2024; Liu et al., 2024b), and their common structures are ill-suited for two-view correspondence learning with sparse points. Although some recent endeavors have explored using Mamba for point cloud tasks (Liu et al., 2024a; Zhang et al., 2024), they predominantly focus on handling 3D sequences and lack guidance on adapting to our 2D data while achieving context compression and ensuring no attention leakage.

Therefore, we propose a Mamba-based correspondence learning method MambaMatch, which is superior in terms of effective data selection. Specifically, the channel-aware Mamba filter (CAMF) is the main part, which can parameterize the state space model (SSM) parameters according to the input data. This allows the network to distinguish inliers and outliers in the high-dimensional latent space and give different focus, which is beneficial to correspondence pruning, especially with a high percentage of outliers. Although Mamba is globally aware, its lack of channel awareness (Guo et al., 2024) makes it underpowered for learning high-dimensional features with many channels. This may lead to the model being sensitive to randomly distributed outliers. To this end, we customize its channel awareness capability so that our method can better adapt to real-world scenes. Meanwhile, it's known that a fundamental principle in correspondence learning is the ability to capture local context (Zhang et al., 2019; Liu et al., 2021). Therefore, to increase the capability to obtain context information (Yi et al., 2018) of the initial sequences, we also design a local context enhancement module (LCEM). As shown in Figure 1(d), our Mamba-based approach not only focuses on potentially correct matches with higher weights but also discards false matches, avoiding attention leakage and achieving context compression. To sum up, the contributions of this article mainly include the following three folds:

i. To our best knowledge, this represents the inaugural exploration into leveraging Mamba for sparse feature matching. Additionally, we pioneer the application of Mamba's selection traits to tackle scenarios characterized by a significant presence of outliers adeptly. This approach could potentially offer valuable insights for a variety of other applications;

ii. We specifically engineer a new outlier filter based on Mamba that can more effectively filter out mismatches by discarding particular inputs, trying to retain correct matches only. Concurrently, we develop a local context enhancement module to address Mamba's deficiency in capturing local context, thereby enhancing the learning of correspondences;

iii. We design a suite of experiments to substantiate the rationality of our methodological design and further demonstrate the efficacy and robust generalization capabilities of our approach through various real-world datasets.

## 2 RELATED WORK

### 2.1 CORRESPONDENCE LEARNING

A renowned paradigm involves first obtaining an initial set of Correspondences, followed by the application of outlier rejection to refine and achieve a more accurate correspondence set. The best-

known handcrafted methods such as RANSAC (Fischler & Bolles, 1981) and its variants (Torr & Zisserman, 2000; Ni et al., 2009; Chum & Matas, 2005) employ a hypothesis-verification strategy to find a maximal consistent subset that fits a particular geometric model. This type of method is highly dependent on the sampled subsets, leading to their failure in high outlier situations. Methods for nonparametric models (Ma et al., 2014; Bian et al., 2017; Fan et al., 2023) can handle both rigid and non-rigid deformations. But they are equally helpless in the face of large perspectives and repetitively structured real-life scenarios. Then, with the development of deep learning, MLPs are first used as a solution for correspondence pruning. Yi et al. (2018) utilizes them to extract high-dimensional features of each putative correspondence individually and introduces context normalization (CN) to capture global contextual information. Zhang et al. (2019) introduces a differentiable layer that captures local information by softly assigning nodes to a set of clusters. Liu et al. (2021) combines global and local coherence to robustly detect true correspondences. Zhang & Ma (2023) uses a CNN as the backbone and avoids the design of additional context normalization modules of MLP-based approaches. However, these methods treat each input equally and lack the ability to mine the different information between inliers and outliers. Some subsequent approaches like NCMNet (Liu & Yang, 2023) and MC-Net (Li et al., 2024) attempt to address the divergence of information between outliers and inliers by learning an attention map that softly assigns varying degrees of focus to different regions. However, they suffer from attention leakage to outliers and efficiency issues due to the inability to compress context.

Another popular pipeline involves directly obtaining an accurate set of Correspondences in one go. Representative work such as SuperGlue (Sarlin et al., 2020) has made significant strides by utilizing graph neural networks in conjunction with attention mechanisms (Vaswani et al., 2017). However, these methods experience a quadratic decrease in efficiency as the number of keypoints increases, leading to difficulties in practical application. Detector-free dense matching methods (Edstedt et al., 2023; 2024) and semi-dense approaches (Sun et al., 2021; Tang et al., 2022)can achieve accurate matches even in extreme scenarios like textureless regions, but the increased computational cost and memory usage due to richer matches remain unresolved. Therefore, our paper focuses on the first paradigm, while the second pipeline is discussed appropriately in the analysis.

## 2.2 STATE SPACE MODELS

The state space model (SSM) is initially used to describe dynamic systems. It has recently been introduced as a generalized backbone for natural language processing and computer vision (Gu et al., 2021b;a; Smith et al., 2022). The problem of early SSMs is similar to RNN models that easily forget global contextual information and suffer from gradient vanishing as the sequence length grows (Gu et al., 2020; 2021b). The recent work Mamba (Gu & Dao, 2023) based on SSM successfully overcomes these shortcomings. With the subquadratic complexity and selective scanning mechanism, Mamba (Gu & Dao, 2023) has the potential to be a prospective backbone. Zhu et al. (2024) and Liu et al. (2024b) innovatively introduce Mamba to vision tasks. In addition, Liu et al. (2024a) and Zhang et al. (2024) first propose to use Mamba to process 3D sequence data. The methods directly employed for handling image data may not directly translate to our task because of the difference in data structure. At the same time, approaches for point cloud tend to overly focus on data preprocessing without significantly enhancing Mamba itself, more resembling a custom utilization. We observe that while Mamba exhibits strong capability in capturing global information, it falls short in channel information acquisition (Guo et al., 2024). Therefore, in this article, while we use Mamba to solve the problem of how to selectively mine and utilize inlier information while discarding outliers, we design the channel-aware module to increase its ability to acquire cross-channel information. This may also offer a novel approach to addressing other scenarios with a high proportion of outliers.

## 3 PRELIMINARIES

### 3.1 REVISITING MAMBA

Mamba (Gu & Dao, 2023) is originated from state space models (SSMs), which are initially employed for linear time-invariant systems to map the input $x(t) \in \mathbb{R}^L$ to the output $y(t) \in \mathbb{R}^L$ (Kalman,

1960), it can be discretized to a discrete-time SSM by zero-order hold (ZOH) discretization:

$$\dot{h}(t) = Ah(t) + Bx(t),$$
$$y(t) = Ch(t), \tag{1}$$

where $h(t) \in \mathbb{R}^N$ is the hidden state, $\dot{h}(t) \in \mathbb{R}^N$ is the derivative of the hidden state. $A \in \mathbb{R}^{N \times N}$, $B \in \mathbb{R}^{N \times L}$, and $C \in \mathbb{R}^{L \times N}$ are the parameters of the model. The continuous-time SSM can be discretized to a discrete-time SSM (Gu & Dao, 2023) by zero-order hold (ZOH) discretization as:

$$h_k = \bar{A}h_{k-1} + \bar{B}x_k,$$
$$y_k = \bar{C}h_k, \tag{2}$$

where $\bar{A}$, $\bar{B}$ and $\bar{C}$ are discrete forms of $A$, $B$, and $C$. By iterating Equation 2, the model output $y$ manifests as the convolution between the input $x$ and a kernel. This characteristic endows Mamba with the inherent capability for parallel computation, illustrated succinctly as:

$$y = \text{SSM}(\bar{K}, x) = \bar{K} * x, \tag{3}$$

where $\text{SSM}(\cdot)$ means the original state space model, and $\bar{K}$ is the convolution kernel represented by:

$$\bar{K} = (C\bar{B}, C\bar{A}\bar{B}, ..., C\bar{A}^{M-1}\bar{B}), \tag{4}$$

where $M$ denotes the sequence length of $x$. Moreover, Mamba can incorporate a selection mechanism that allows the parameters to be changed from fixed to a function of the inputs, while changing the tensor shape. We use linear layers denoted as:

$$s_B(x) = \text{Linear}(x), s_C(x) = \text{Linear}(x), \tag{5}$$

where $s_B(x)$ and $s_C(x)$ are the parameter matrixs $B$ and $C$ with regard to the input $x$, respectively.

## 3.2 PROBLEM FORMULATION

Given an image pair $(\mathbf{I}, \mathbf{I}')$, we employ off-the-shelf feature detectors and descriptors to extract keypoints from both images. Subsequently, employing the nearest neighbor (NN) method, we establish initial putative matches. Denoted as $\mathbf{C} = [\mathbf{c}_1; \mathbf{c}_2; \cdots]$, where $\mathbf{c}_i = \{(x_i, y_i, x'_i, y'_i)|i = 1, \cdots, N\}$, $(x_i, y_i)$ and $(x'_i, y'_i)$ represent the coordinates of the $i$-th keypoints in respective images.

In analogy to Yi et al. (2018), we approach the two-view correspondence learning problem as an inlier/outlier classification and an essential matrix regression. For the putative correspondences $\mathbf{C} \in \mathbb{R}^{N \times 4}$, to extract the deep information, they are usually upscaled to get $\mathbf{F} = \{\mathbf{f}_1, \mathbf{f}_2, \cdots, \mathbf{f}_i\} \in \mathbb{R}^{N \times d}$. Moreover, we establish an inlier predictor at each layer of the network for simultaneous training, with only the last predictor yielding the probability value $\mathbf{P} = [p_1, p_2, \cdots, p_N]^T \in \mathbb{R}^{N \times 1}$, where $p_i \in [0, 1)$ signifies the probability that the corresponding $\mathbf{c}_i$ is an inlier. Like other learning-based methods (Liu et al., 2021; Liu & Yang, 2023; Li et al., 2024), we employ a weighted eight-point method based on $\mathbf{P}$ to directly estimate the essential matrix. The entire process is encapsulated as follows:

$$\widehat{\mathbf{P}} = f_\phi(\mathbf{C}), \quad \widehat{\mathbf{E}} = g(\widehat{\mathbf{P}}, \mathbf{C}), \tag{6}$$

where $f_\phi(\cdot)$ undertakes inlier prediction, and $g(\cdot)$ signifies the estimation of the parametric model.

## 4 METHODOLOGY

In order to adaptively learn to mine the meaningful information of inliers and discard the redundant counterpart of outliers, we introduce a mamba-inspired network named MambaMatch. As depicted in Figure 2, our framework initiates with the Position-Context Initialization (see Figure 2(b)) for feature upscaling, proceeds with LCEM (see Figure 2(c)) for learning local context and CAMF for selectively mining useful information, and culminates in the Cluster-Sequence Block as well as Inlier Predictor (see Figure 2(d)) to obtain inlier probabilities. Drawing on the works of Zhang et al. (2019) and Liu & Yang (2023), we combine the Cluster-Sequence Block in our framework without delving into intricate specifics. In addition, we will describe CAMF in detail and give a specific structure in Section 4.1. Notably, the MambaMatch layer (see Figure 2(a)), a fusion of LCEM, CAMF, and Cluster-Sequence Block, constitutes the stacked layers denoted by $L$ (*e.g.*, $L = 4$ in our implementation).

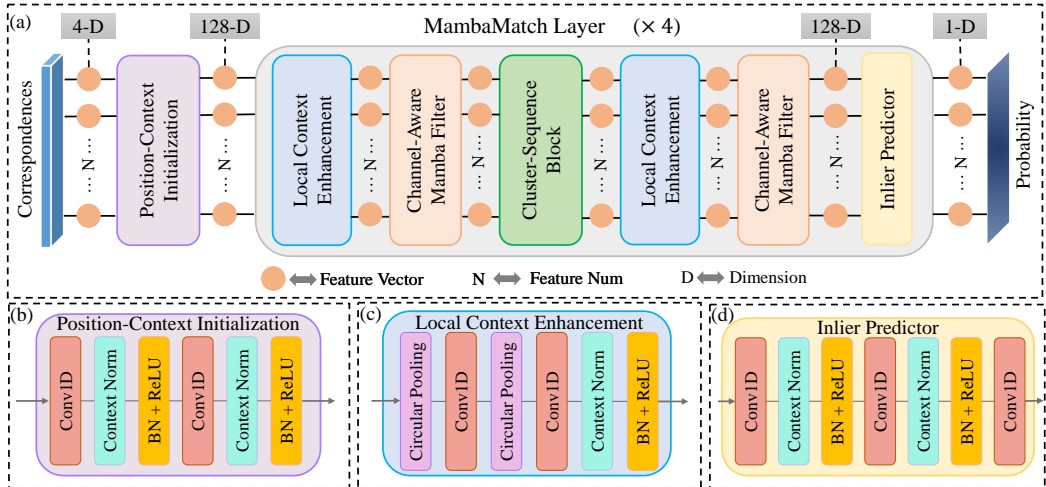

Figure 2: Framework diagram of MambaMatch. We use the putative set of correspondences obtained by off-the-shelf descriptors and detectors as input and finally obtain the inlier probability of each correspondence through the network.

## 4.1 CHANNEL-AWARE MAMBA FILTER

As mentioned earlier, the significance of the information provided by inliers and outliers varies. Strategically mining and effectively utilizing the insights from these diverse points are crucial for corresponding pruning. To this end, leveraging Mamba's inherent selection traits, we design a Mamba-based filter. This enables the network to inherit Mamba's characteristics, allowing it to focus on the inputs from inliers while disregarding outlier information and compressing the deep features into more concise representations. First of all, we can rewrite Equation 3 as:

$$\mathbf{F}_M = \text{SSM}(\bar{K}_F, \mathbf{F}_{LC}) = \bar{K}_F * \mathbf{F}_{LC}, \tag{7}$$

where $\mathbf{F}_{LC}$ denotes the input of this module, *i.e.*, the enhanced feature obtained from the original input $\mathbf{F}$ after LCEM, which will be introduced concretely in Section 4.2, and $\bar{K}_F$ means the convolutional kernel. To further realize the selectivity as expressed in Equation 5, we perform this function using the left part of Figure 3. In conclusion, it can be delineated as follows:

$$\mathbf{F}_M = \mathbf{F}_{LC} + \mathcal{M}(\text{SSM}(\text{CNN}(\mathcal{M}(\mathbf{F}_{LC}))) \odot \delta(\mathcal{M}(\mathbf{F}_{LC}))), \tag{8}$$

where $\mathcal{M}(\cdot)$ means MLP, $\text{CNN}(\cdot)$ denotes 1D convolutional neural network, $\odot$ refers to the Hadamard product, *i.e.*, the product between elements, and $\delta(\cdot)$ represents the activation function, *i.e.*, SiLU (Shazeer, 2020).

Therefore, as mentioned above, the proposed Mamba Filter can focus on or discard particular inputs to mining useful information, but it underperforms in cross-channel information access (Guo et al., 2024) while channel-wise information is significant in high-quality correspondence recognition. To enhance the expressive power of different channels, we customarily design a channel-aware module as shown in Figure 3, denoted as:

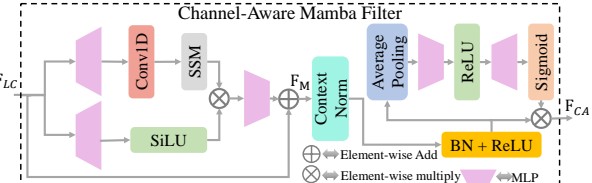

Figure 3: Architecture of Chnnal-Aware Mamba Filter.

$$\mathbf{F}_{CA} = \mathbf{F}_M \odot W, \tag{9}$$

where $W$ represents the expressive capabilities of different channels, we define weights:

$$W = \text{Sigmoid}(\mathcal{M}(\delta(\mathcal{M}(\text{AP}(\mathbf{F}_M))))), \tag{10}$$

where $\delta(\cdot)$ means ReLU activation fuction, while $\text{AP}(\cdot)$ denotes average pooling. Additionally, for chaotic putative matches, it is necessary to supplement channel learning with context information.

Following Yi et al. (2018), we introduce Context Normalization and optimize Equation 9 and Equation 10 as:

$$\mathbf{F}_{CA} = \mathbf{F}_{CN} \odot W',$$
$$\mathbf{F}_{CN} = \delta(\text{BN}(\text{CN}(\mathbf{F}_M))), \tag{11}$$
$$W' = \text{Sigmoid}(\mathcal{M}(\delta(\mathcal{M}(\text{AP}(\mathbf{F}_{CN}))))),$$

where $\text{CN}(\cdot)$ and $\text{BN}(\cdot)$ denotes Context Normalization and Batch Normalization. Through this approach, our Mamba Filter can focus on learning distinct channel representations and subsequently select crucial channels via subsequent channel attention, thereby circumventing channel redundancy.

## 4.2 LOCAL CONTEXT ENHANCEMENT MODULE

Though Mamba (Gu & Dao, 2023) excels in global information acquisition, it's also essential to consider local context learning (Zhang et al., 2019) for two-view learning. To this end, we design a Local Context Enhancement Module (LCEM) to improve the acquisition of local context. In order to take full advantage of the information between local nodes, we construct a neighbor graph $\mathcal{G}_i = \{\mathcal{V}_i, \mathcal{E}_i\}$ in the feature space for each correspondence $\mathbf{f}_i$ based on its spatial adjacency. $\mathcal{V}_i = \{\mathbf{f}_{i1}, \cdots, \mathbf{f}_{ik}\}$ denotes the $k$-nearest neighbors of $\mathbf{f}_i$ and $\mathcal{E}_i = \{e_{i1}, \cdots, e_{ik}\}$ denotes the directed edges connecting the anchor and its neighbors in the feature space. We use an edge construction strategy similar to Zhao et al. (2021):

$$e_{ij} = [\mathbf{f}_i || \mathbf{f}_i - \mathbf{f}_{ij}], j = 1, 2, \cdots, k, \tag{12}$$

where $[\cdot||\cdot]$ denotes the concatenation operation along the channel dimension.

Once the graph of the feature space is constructed, we need to consider how to effectively mine intra-neighborhood consistency. To address the exigencies of real-world scenarios characterized by substantial disparities and sparse matches, we advocate for the adoption of circular pooling, an approach poised to capture the nuanced neighborhood consistency inherent in matches. Concretely, our method entails the initial segmentation of the $k$ nearest neighbors surrounding the anchor into $\frac{k}{p}$ cohorts, where $p$ constitutes a divisor of $k$ and means the number of neighbors in each cohort. Subsequent to two iterations of circular pooling, the resulting formulation is expressed as:

$$\mathbf{F}_{CP} = \text{CP}_2(\text{CP}_1(\mathcal{E}_i)). \tag{13}$$

Here $\text{CP}_i(\cdot)$ denotes the circular pooling, *i.e.*, the convolutions with $1 \times \frac{k}{p}$ kernels and $1 \times p$ respectively. In addition, to incorporate more contextual information, we continue to increase the context-aware capability of circular pooling. Specifically, we perform the context normalization as before on the output of the circular poolings, which can be represented as follows:

$$\mathbf{F}_{LC} = \delta(\text{BN}(\text{CN}(\mathbf{F}_{CP}))), \tag{14}$$

where $\mathbf{F}_{LC}$ is subsequently fed into the CAFM for selective information extraction.

In summary, our initial input feature $\mathbf{F}$ is finalized by LCEM and CAMF to get $\mathbf{F}_{CA}$ with Equation 13, Equation 14, Equation 8 and Equation 11. Next, we use the inlier predictor $\text{IP}(\cdot)$ as shown in Figure 2(d) to predict the probability of getting the match to be an inlier, denoted as follows:

$$\mathbf{p} = \text{IP}(\mathbf{F}_{CA}). \tag{15}$$

## 4.3 LOSS FUNCTION

We choose a widely used loss function (Yi et al., 2018; Zhang & Ma, 2023) as:

$$L = \sum_{l=1}^{\mathcal{L}} \alpha L_{cls}(^{(l)}\mathbf{p}, \mathbf{Z}) + \beta L_{reg}(^{(l)}\widehat{\mathbf{E}}, \mathbf{E}). \tag{16}$$

Herein, $\mathbf{p}$ represents the output probabilities of an inlier predictor as shown in Equation 15, $\mathbf{Z} = \{z_i\}_{i=1}^N$ encapsulates weakly supervised labels derived via geometric error (Hartley & Zisserman, 2003), $\widehat{\mathbf{E}}$ denotes the estimated essential matrix, and $\mathbf{E}$ stands for the ground-truth. $\alpha$ and $\beta$ are adeptly employed to harmonize the contributions of the two loss terms. $L_{cls}(\cdot)$ denotes a rudimentary

binary cross-entropy loss designed for the classification aspect, while $L_{reg}(\cdot)$ is ascertained utilizing the Sampson distance (Hartley & Zisserman, 2003):

$$L_{\mathrm{reg}}\left(\widehat{\mathbf{E}}, \mathbf{E}\right) = \sum_{i=1}^{N} \frac{\left(\mathbf{t}_i'^{\top}\widehat{\mathbf{E}}\mathbf{t}_i\right)^2}{\|\mathbf{E}\mathbf{t}_i\|_{[1]}^2 + \|\mathbf{E}\mathbf{t}_i\|_{[2]}^2 + \|\mathbf{E}^T\mathbf{t}_i'\|_{[1]}^2 + \|\mathbf{E}^T\mathbf{t}_i'\|_{[2]}^2}, \tag{17}$$

where $\mathbf{t}_i$ and $\mathbf{t}_i'$ represent two keypoints that constitute the correspondence $\mathbf{c_i}$, and $||v||_{[i]}$ denotes the $i$-th element of vector $\mathbf{v}$.

### 4.4 IMPLEMENTATION DETAILS

For implementation, we normalize the coordinates of keypoints to [-1, 1] with image size or camera intrinsic. Our MambaMatch consists of $L = 4$ stacked MambaMatch layers. The geometric error threshold is set to $10^{-4}$. The model is optimized by Adam (Kingma & Ba, 2022), and the learning rate is set to $10^{-3}$ during the first $80k$ iterations then decaying with a factor of $0.999996$ every step. We use a batch size of $32$, with weights $\beta$ starting at $0$ and then $0.5$ after the first $20k$ iterations, while $\alpha$ is fixed at $1$ throughout the training process. Training is terminated after $700k$ iterations. All training and testing are performed with a single RTX3090 GPU.

## 5 EXPERIMENT

### 5.1 RELATIVE POSE ESTIMATION

**Datasets.** In experiments for relative pose estimation, we choose the YFCC100M dataset (Thomee et al., 2016) to demonstrate our method's capability to learn in outdoor environments and the SUN3D dataset (Xiao et al., 2013) to showcase its performance in indoor settings. The YFCC100M dataset consists of 100 million outdoor images collected from the web and is divided into 72 distinct sequences. In line with previous research (Yi et al., 2018), we allocate 68 sequences for training and validation

Table 1: Results (AUC@5°/@10°/@20°) of relative pose estimation. The best results are marked in bold.

| Method | YFCC100M | SUN3D |
|---|---|---|
| GMS | 13.29/24.38/37.83 | 4.12/10.53/20.82 |
| LPM | 15.99/28.25/41.76 | 4.80/12.28/23.77 |
| CRC | 16.51/28.01/41.38 | 4.07/10.44/20.87 |
| VFC | 17.43/29.98/43.00 | 5.26/13.05/24.84 |
| PointCN | 26.73/44.01/60.49 | 6.09/15.43/29.74 |
| OANet | 27.26/45.93/63.17 | 6.78/17.10/32.41 |
| CLNet | 31.45/51.06/68.40 | 6.67/16.81/31.45 |
| ConvMatch | 31.69/51.41/68.45 | 7.32/18.45/34.41 |
| NCMNet | 32.30/52.29/69.65 | 7.10/18.56/**35.58** |
| MC-Net | 33.02/52.42/69.23 | 7.40/18.72/34.81 |
| MambaMatch (Ours) | **33.48/53.23/70.48** | **7.67/18.87**/34.87 |

purposes, while the remaining 4 sequences are earmarked for testing. As for the SUN3D dataset, it contains original indoor RGBD video frames, from which we sample every 10th frame. We select 239 sequences for training and validation, following the testing protocol established by other methods (Zhang et al., 2019; Li et al., 2024), which reserves 15 sequences exclusively for testing.

**Evaluation Protocols.** We assess the accuracy of pose estimation by analyzing the area under the cumulative error curve (*i.e.*, AUC) for pose errors across various thresholds (5°, 10°, 20°). Pose error is defined as the maximum of the angular error in rotation and translation. We use SIFT (Lowe, 2004) to extract up to $2k$ keypoints and acquire putative matches with the NN method.

**Baseline.** In our experiments, we categorize the approaches based on their underlying principles and endeavor to compare a wide array of the SOTA, encompassing traditional approaches (GMS (Bian et al., 2017), LPM (Ma et al., 2019), CRC (Fan et al., 2023), VFC (Ma et al., 2014)) and learning-based techniques (PointCN (Yi et al., 2018), OANet (Zhang et al., 2019), CLNet (Zhao et al., 2021), ConvMatch (Zhang & Ma, 2023), NCMNet (Liu & Yang, 2023), MC-Net (Li et al., 2024)).

**Results.** Table 1 presents the estimation outcomes for both the YFCC100M dataset and the SUN3D dataset. Drawing upon pertinent literature (Sarlin et al., 2020; Zhao et al., 2021), we adopt RANSAC (Fischler & Bolles, 1981) as our robust essential matrix estimator. Employing

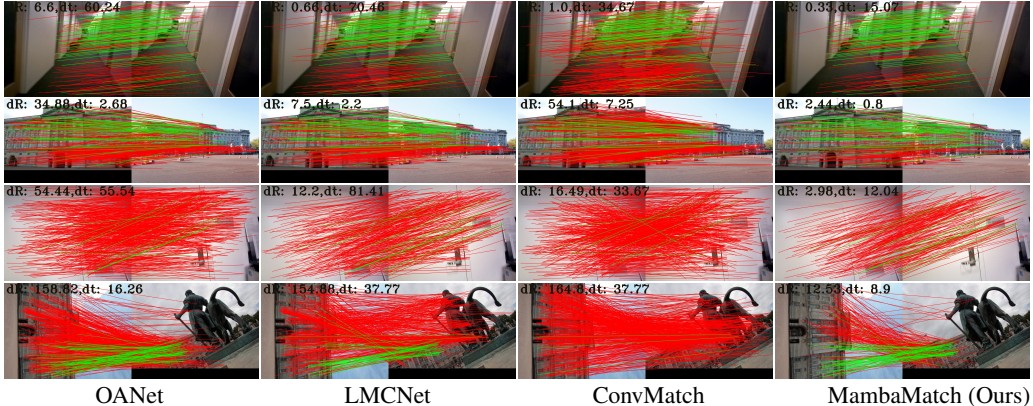

OANet      LMCNet      ConvMatch      MambaMatch (Ours)

Figure 4: Qualitative illustration of outlier rejection. False matches are marked with red (—) while correct matches are with green (—). The relative pose estimation results (error of rotation and translation) are provided in the top left corner of each image pair. Please zoom in for a better view.

the innovative Mamba architecture for feature matching, we juxtapose our proposed MambaMatch against popular methodologies. It is evident that our method surpasses nearly all qualitative assessments compared to the previous SOTA methods, with the closest competitor significantly trailing behind us in terms of time efficiency (refer to Section 5.3). Additionally, we provide qualitative insights into outlier rejection, as illustrated in Figure 4. Our approach excels at preserving inliers and effectively eliminating outliers, thereby achieving relative pose estimation with minimal rotational and translational errors. We present additional visualizations in the *Appendix*.

## 5.2 VISUAL LOCALIZATION

To substantiate the practical applicability of our approach, we conduct experiments on visual localization using the official HLoc pipeline (Sarlin et al., 2019). Engineered to pinpoint the 6-degree-of-freedom (6-DOF) orientation of query images within a 3D architectural context, this framework serves as the foundation for our assessment. Building upon it, we scrutinize the proficiency of our technique in yielding robust matching outcomes under demanding conditions, including shifts in viewpoint and transitions from daytime to nocturnal illumination.

Table 2: Visual localization results.

| Method | Day | Night |
|---|---|---|
| | (0.25m,2°)/(0.5m,5°)/(1.0m,10°) | |
| PointCN | 83.1/92.2/96.2 | 69.4/79.6/89.8 |
| OANet | 83.3/92.5/96.6 | 71.4/80.6/90.8 |
| CLNet | 83.3/92.4/97.0 | 71.4/80.6/**93.9** |
| MS$^2$DGNet | 84.2/92.8/97.0 | **74.5/83.7**/91.8 |
| LMCNet | 84.1/92.8/**97.1** | 71.4/81.6/**93.9** |
| ConvMatch | 84.5/92.7/96.8 | 73.5/83.7/91.8 |
| MambaMatch (Ours) | **85.1/93.0/97.1** | 72.4/**83.7/93.9** |

**Datasets.** Skin to Li et al. (2024), we leverage the well-established HLoc pipeline to assess the effectiveness of our methodology in visual localization using the Aachen day-night dataset (Sattler et al., 2018). This dataset encompasses 4328 images capturing Aachen city, accompanied by 922 query images, comprising 824 daytime and 98 nighttime snapshots, all meticulously captured by smartphone cameras.

**Evaluation Protocols.** As per the authoritative assessments (Sarlin et al., 2019), we present the proportion of accurately localized queries within specified thresholds of distance and orientation. It's noteworthy that we employ SIFT (Lowe, 2004) to extract up to $4,096$ keypoints from each image. These keypoints are subsequently matched utilizing the NN to establish putative correspondences. Following this, triangulation is performed on the SfM model using daytime images with known poses. Finally, we leverage correspondence learning for 2D matching and register nighttime query images with the COLMAP framework (Schonberger & Frahm, 2016).

**Results.** Table 2 shows the results of visual localization. MambaMatch achieves the best results in most conditions both daytime and nighttime. Additionally, the qualitative results are shown in

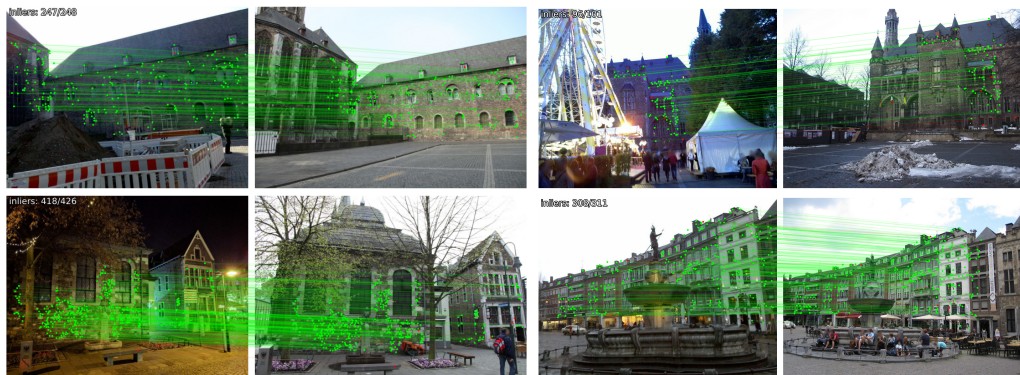

Figure 5: Visualization of visual localization. Please zoom in for a better view.

Figure 5. Red dots represent outliers, while green lines connect inliers. The top left corner displays the proportion of inliers among all detected points. Apparently, our method is able to accurately locate the correct match for scenes with large viewpoint changes, occlusion, and lighting changes.

## 5.3 ANALYSIS

**Computational Usage.** The previous experiments demonstrate the effectiveness of our approach on various visual tasks. To further validate the efficiency of our method, we evaluate several performance metrics, including model parameters (P.), floating-point operations (Flops.), and average running time per image (T.). These evaluations are conducted using the same test dataset ($4k$ images, each extracted $2k$ keypoints using SIFT (Lowe, 2004)). The results, presented in Table 3, indicate that our method offers advantages over vanilla attention-based methods. While LMCNet (Liu et al., 2021) significantly reduces model size by leveraging additional library functions, its time consumption is relatively high due to graph construction and matrix decomposition used to solve explicit regularization terms.

Table 3: Computational usage.

| Method | P. (M) | Flops. (G) | T. (ms) |
|---|---|---|---|
| LMCNet | 0.925 | - | 227.15 |
| NCMNet | 4.485 | 8.717 | 138.36 |
| MC-Net | 50.393 | 8.720 | 49.40 |
| MambaMatch (Ours) | 3.476 | 7.810 | 36.45 |

**Compatibility with Matchers.** In the experiments described above, we thoroughly showcase the outstanding capabilities of MambaMatch when utilized with SIFT (Lowe, 2004) and the NN matchers. Serving as a versatile backend outlier filter, we assessed the performance of MambaMatch on YFCC100M (Thomee et al., 2016) for relative pose estimation, with or without various widely adopted matchers. These include Super-Point (DeTone et al., 2018) paired with SuperGlue (Sarlin et al., 2020) (referred to as SP + SG), SuperPoint paired with LightGlue (Lindenberger et al., 2023)

Table 4: Compatibility with Matchers.

| Matcher | Filter | Estimator | @5° | @10° | @20° |
|---|---|---|---|---|---|
| SP + SG | ✗ | RANSAC | 38.06 | 58.38 | 74.67 |
| | ✓ | | 38.58 | 58.92 | 75.13 |
| | ✗ | PROSAC | 39.41 | 59.00 | 74.91 |
| | ✓ | | 40.19 | 59.48 | 75.81 |
| SP+LG | ✗ | RANSAC | 39.42 | 59.69 | 75.89 |
| | ✓ | | 39.63 | 59.81 | 76.02 |
| | ✗ | PROSAC | 39.93 | 59.82 | 74.21 |
| | ✓ | | 40.26 | 59.95 | 74.97 |
| LoFTR | ✗ | RANSAC | 39.84 | 60.51 | 76.42 |
| | ✓ | | 41.45 | 62.06 | 77.64 |
| | ✗ | PROSAC | 42.58 | 58.14 | 70.04 |
| | ✓ | | 42.65 | 59.24 | 71.70 |

(referred to as SP + LG), and LoFTR (Sun et al., 2021), while employing RANSAC (Fischler & Bolles, 1981) or PROSAC (Chum & Matas, 2005)for pose estimation. For SP + SG and SP + LG, we adhere to the settings of SuperGlue and detect up to $2,048$ keypoints. Evaluation procedures for LoFTR closely follow those outlined in Truong et al. (2021). It's worth noting that, in line with recommendations from similar experiments in Liu et al. (2021), we refrain from employing a filtering strategy in each method, opting instead to retain all putative correspondences as inputs. The findings presented in Table 4 demonstrate that as a generalized outlier filtering approach, MambaMatch

Table 5: Generalization ability test. We report the AUC@5 metric, highlighting the best results in bold and the second-best results with underlining.

| Method | YFCC100M | | | | SUN3D | | | |
|---|---|---|---|---|---|---|---|---|
| | SIFT | RootSIFT | LIFT | SuperPoint | SIFT | RootSIFT | LIFT | SuperPoint |
| PointCN | 26.73 | 27.33 | 19.06 | 24.74 | 5.89 | 6.06 | 5.10 | 5.31 |
| OANet | 27.26 | 30.05 | 21.54 | 26.31 | 5.42 | 5.83 | 5.07 | 5.21 |
| CLNet | 31.45 | 32.42 | 22.48 | 27.44 | **6.39** | **6.52** | 4.85 | 3.37 |
| LMCNet | 30.48 | 31.65 | 23.36 | 26.77 | 5.71 | 5.98 | 5.40 | 5.52 |
| ConvMatch | 31.69 | 33.05 | 24.38 | **29.43** | 5.96 | 6.26 | 5.40 | 5.56 |
| MambaMatch (Ours) | **33.48** | **34.44** | **25.45** | 29.39 | 6.16 | 6.30 | **5.58** | **6.61** |

consistently enhances advanced matchers and can serve as a complementary module in practical applications.

**Generalization Ability.** To evaluate the performance of MambaMatch in various scenarios with descriptors, we first trained several learning-based methods using SIFT (Lowe, 2004) on YFCC100M (Thomee et al., 2016). We then test these models on YFCC100M with RootSIFT (Arandjelović & Zisserman, 2012), LIFT (Yi et al., 2016), and SuperPoint (DeTone et al., 2018), as well as on SUN3D (Xiao et al., 2013) using SIFT, RootSIFT, LIFT, and SuperPoint for pose estimation. For each image, we extracted up to $2k$ keypoints using different descriptors, with putative matches generated using the NN method. As shown in Table 5, MambaMatch consistently achieved the best results in almost all cases, clearly demonstrating its superior generalization ability.

**Ablation Studies.** We conduct ablation studies by repeatedly performing relative pose estimation. The results are presented in Table 6. Here, LCEM means our proposed Local Context Enhancement Module, Mamba refers to the vanilla Mamba Filter we introduced, and C.A. indicates whether a channel-aware design is applied to the Mamba or not. We report the AUC@5°using RANSAC as an estimator on YFCC100M (Thomee

Table 6: Results of ablation studies.

| N | LCEM | CAMF | | F.F. | L.F. | @5° |
|---|---|---|---|---|---|---|
| | | Mamba | C.A. | | | |
| i. | ✓ | | | | | 30.60 |
| ii. | ✓ | ✓ | | | | 30.64 |
| iii. | ✓ | | ✓ | | | 30.22 |
| iv. | | ✓ | ✓ | | | 32.53 |
| v. | ✓ | ✓ | ✓ | | | 33.48 |
| vi. | ✓ | | ✓ | ✓ | | 31.84 |
| vii. | ✓ | | ✓ | | ✓ | 31.36 |

et al., 2016). As shown in i) and ii), ordinary Mamba has a limited increase in modeling capability, while C.A. in iv) makes for a larger performance improvement. And comparisons between iv) and v) can demonstrate the positive contribution of capturing local context. The best results are achieved when the full set of our proposed modules is utilized. In addition to the ablation of the fundamental structure, we also explored the substitution of our Mamba Filter with alternative attention mechanisms, as illustrated in vi) and vii). Both FastFormer (F.F.) (Wu et al., 2021) and LinFormer (L.F.) (Wang et al., 2020) are found to be slightly inferior to our approach.

# 6 CONCLUSION AND PROSPECTS

In this paper, we design a novel network named MambaMatch for two-view correspondence learning. Inspired by the selective state space model, MambaMatch can focus on or discard particular inputs. Specifically, by targeting the distinguishability of inliers and outliers in the high-dimensional hidden space, the network gives different focus to the two, so that it can pay more focus to the useful information brought by the inliers while discarding the interfering information of the outliers to realize high-precision correspondence pruning. Meanwhile, since ignoring irrelevant information will compress the context at the same time, resulting in redundant information being discarded directly, this makes the model have subquadratic complexity. This is relatively friendly for handling real-time tasks. A large number of experiments prove the superior performance of our method.

Additionally, our approach, which focuses on the selection characteristics of Mamba, may offer insights for a range of tasks with a high proportion of noisy information interference.

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

# A APPENDIX

## A.1 QUALITATIVE RESULTS FOR RELATIVE POSE ESTIMATION

We show more visualization results (including OANet Zhang et al. (2019), LMCNet Liu et al. (2021), ConvMatch Zhang & Ma (2023) and MambaMatch) of outlier rejection and relative pose estimation for outdoor scenes (the 1-st row to the 6-th row) and indoor scenes (the 7-th row to the 10-th row) in Figure 6. Note that our MambaMatch is able to handle more complex scenes of large viewing angle deviation with good results.

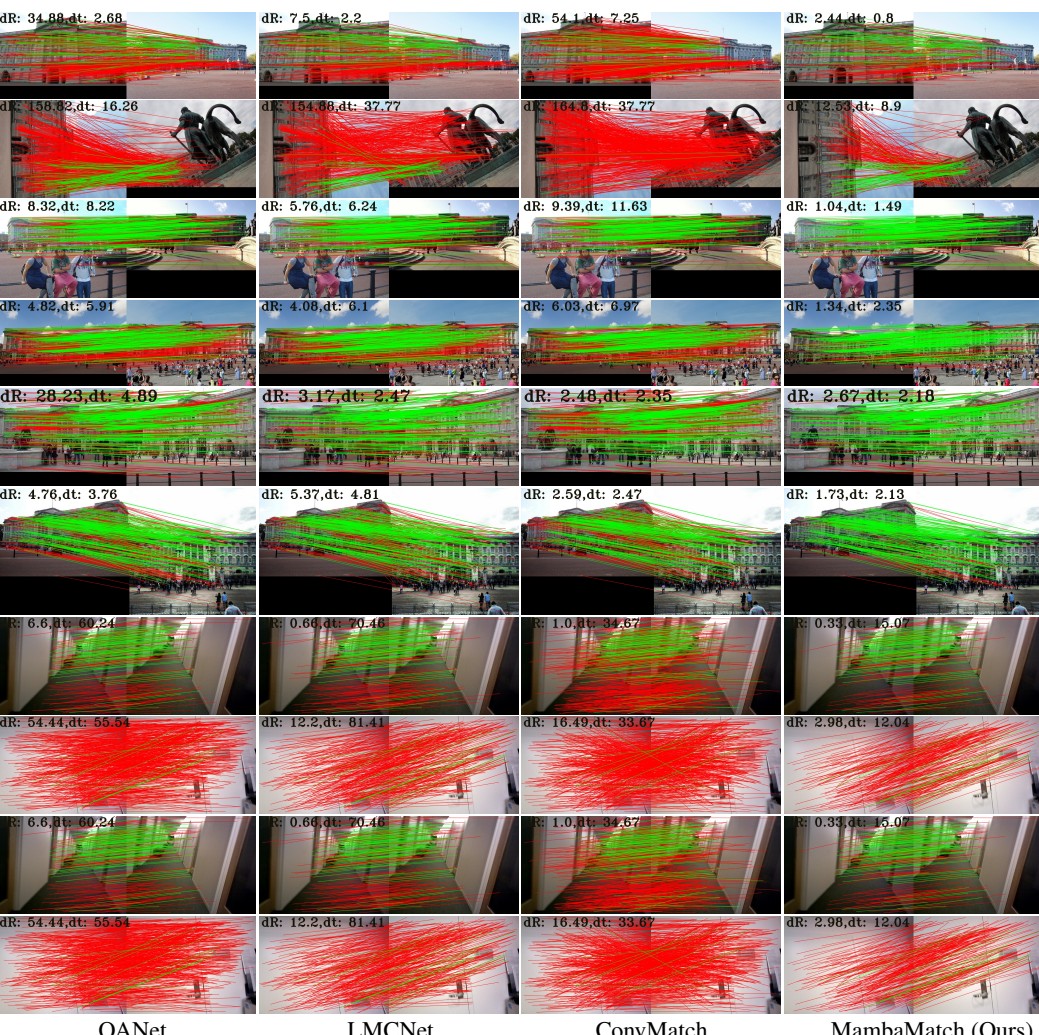

| OANet | LMCNet | ConvMatch | MambaMatch (Ours) |

Figure 6: Qualitative illustration of outlier rejection. False matches are marked with red (—) while correct matches are with green (—). The relative pose estimation results (error of rotation and translation) are provided in the top left corner of each image pair. Please zoom in for a better view.

## A.2 LIMITATIONS

Although our method has outperformed most attention-based methods in terms of efficiency, it perhaps leaves much to be desired in terms of time and space when compared to MLP-based methods.

In addition, compared with powerful dense matching methods, our method may also fall short in performance.

