# OpenReview forum: "MambaMatch: Learning Two-View Correspondences with Selective State Spaces"
_ICLR.cc/2025/Conference — ICLR 2025 Conference Withdrawn Submission_

### Official Review · Reviewer_xce1 · 2024-11-03

**Soundness:** 3
**Presentation:** 2
**Contribution:** 2
**Rating:** 5
**Confidence:** 3

**Summary:**

Inspired by Mamba’s inherent competence of selectivity, this paper propose MambaMatch as a Mamba-based correspondence filter to selectively mine information from true correspondences and to dispose of the potentially interfering information of false correspondences. Specifically, the selection is achieved by adaptively adjusting model parameters in a high-dimensional latent space, which also avoids attention leakage and implements context compression, ensuring the precise and efficient exploitation of pertinent information. Meanwhile, channel awareness is tailored to serve as a complementary aspect of comprehensive information acquisition. Moreover, this paper design a novel local-context enhancement module to capture reasonable local context that is crucial for correspondence pruning. Extensive experiments demonstrate that our approach outperforms existing state-of-the-art methods on several visual tasks while saving time and space costs.

**Strengths:**

The strengths can be as follows:
1. this papers represents the inaugural exploration into leveraging Mamba for sparse feature matching. Additionally, the paper pioneer the application of Mamba’s selection traits to tackle scenarios characterized by a significant presence of outliers adeptly. This approach could
potentially offer valuable insights for a variety of other applications;
ii. this paper specifically engineer a new outlier filter based on Mamba that can more effectively filter out mismatches by discarding particular inputs, trying to retain correct matches only. Concurrently, it develop a local context enhancement module to address Mamba’s deficiency in capturing local context, thereby enhancing the learning of correspondences;
iii. this paper design a suite of experiments to substantiate the rationality of our methodological design and further demonstrate the efficacy and robust generalization capabilities of our approach through various real-world datasets.

**Weaknesses:**

1. For the Computational Usage, it's better to compare the proposed method with latest work "Efficient LoFTR: Semi-Dense Local Feature Matching with Sparse-Like Speed" which shows quite good efficiency and quality.
2. For the "RELATIVE POSE ESTIMATION" experiments, why not use popular datasets like MegaDepth and scanet to compare different methods so that make the paper more solid.
3. For the "VISUAL LOCALIZATION", is it possible to compare the proposed method with LoFTR series and AspanFormer papers?
4. The paper looks like just using the mamba instead of traditional CNN and transformer architexture, though the experiments on the paper show the SOTA result but there are a lot of good works for feature matching is involved.

**Questions:**

No

---

### Official Review · Reviewer_Lypq · 2024-11-04

**Soundness:** 3
**Presentation:** 3
**Contribution:** 2
**Rating:** 5
**Confidence:** 4

**Summary:**

The paper introduces MambaMatch, a Mamba-based correspondence learning method with a focus on effective data selection. Specifically, the Channel-Aware Mamba Filter (CAMF), parametrizes SSM parameters to distinguish inliers and outliers in high-dimensional latent space, especially in outlier-heavy scenarios. To address Mamba’s limitations, the authors enhance channel-awareness and add a Local Context Enhancement Module (LCEM) for improved local context capture. Contributions include pioneering Mamba for sparse feature matching, developing a custom outlier filter and LCEM, and validating MambaMatch’s effectiveness on real-world datasets.

**Strengths:**

1. The paper is well-written and easy to follow, making the methodology and results accessible to the reader.
2. This work inaugurally explore well-known Mamba which key component is SSM to solve key-point correspindence of paired-view images.
3. The proposed method, MambaMatch, demonstrates superior performance over other methods on two benchmark datasets, underscoring its effectiveness in real-world scenarios.

**Weaknesses:**

1. The motivation for the proposed approach is unclear. Since correspondence filtering inherently aims to retain correct matches and discard incorrect ones, the novelty of MambaMatch should be clarified. Specifically, it would help to explain how MambaMatch’s selective capabilities address limitations in previous methods that lack an effective mechanism for prioritizing true correspondences over false ones.
2. The claim that the Mamba-based approach focuses on correct matches, discards false ones, and prevents attention leakage is interesting. However, the paper would benefit from a clearer explanation of how the Mamba structure achieves these capabilities.
3. The comparative experiments are insufficient to fully validate the effectiveness of the proposed approach. For example, the authors claim the use of a specific loss function; however, there is a lack of comparative experiments to demonstrate its effectiveness.

**Questions:**

1. Clarify how to upscale C to F in the section of problem formation as well as what value is for d dimensional.
2. Authors declared selective state space models can ensure the preciseness and efficiency of exploitintg pertinent information. Complexity of the methods should be analyzed to illustrate the greatness of the proposed methods in computational overhead.
3. What is AUC@5 at line 485 in the caption of Table 5.
4. The Mamba does not need to be verbosely introduced in this paper since it is a well-known method that is capable of handling both continuous and discretized problems.
5. Not all SSMs (e.g., the structural SMM) have selectivity and thus Section 2.2 should give more statements for selective SSMs rather than rough SSMs.

---

### Official Review · Reviewer_BQWp · 2024-11-04

**Soundness:** 2
**Presentation:** 2
**Contribution:** 2
**Rating:** 3
**Confidence:** 4

**Summary:**

This paper proposes MambaMatch, a new method building on the efficacy of Mamba networks for two-view correspondences.
The rationale behind this approach is to selectively mine information from true correspondences - relying on the selectivity of Mamba networks - and to dispose the potentially erroneous information from false correspondences.
Instead of simply relying on Mamba networks, the authors proposes to improve the channel-awareness to alleviate the weak cross-channel awareness in Mamba networks.
The authors also propose to capture reasonable local context of correspondences for improved pruning, via the Local-Context Enhancement Module (LCEM).
State-of-the-art experimental results on the tasks of relative pose estimation and visual localization demonstrates the strength of MambaMatch, while incurring lower FLOPs and latency (i.e., more efficient) compared to existing methods.

**Strengths:**

- This paper presents an original idea of integrating Mamba networks for the task of establishing two-view correspondence.
- The authors identify that vanilla Mamba networks lack channel awareness, and integrate a channel-aware module to boost the performance of MambaMatch.
- MambaMatch demonstrates state-of-the-art performances on relative pose estimation and visual localization.
- Additional experimental analysis, such as the generalization experiments and the ablation experiments, provide additional insights into the performances and abilities of MambaMatch.

**Weaknesses:**

## Misleading / Inaccurate presentation / theory

While the overall idea and direction of research is original and interesting, I find that the way it is presented in the paper is severely misleading in multiple aspects.

**1. In Figure 1**, MLP-based filters are shown to output all correspondence weights as 'yellow', which means that all correspondences are seen as vague. I understand that the authors were trying to illustrate that "MLPs treat all input equally (L053)", as MLP weights are static and are fixed regardless of the input at inference. However, this does not mean that all inlier probabilities are predicted to the same value, as static weights still result in different results for different inputs.

Also, while it is understandable that the authors tried to differentiate attention-based filters with Mamba-based filters by illustrating the Mamba-based filters can 'reject' outliers by removing all the red(-ish) lines, Mamba networks do not inherently 'reject' outliers. If a certain type of thresholding was applied to reject correspondences with low inlier ratios, the same can be done for Attention-based methods and MLP-based methods. Please correct me if I have wrongly understood the figure and the authors' intentions.

**2. Resuming on the aspect of 'outlier rejection'**, the authors mention similar ideas throughout the paper, e.g., "dispose of the potentially interfering information of false correspondences (L016)", " enabling models to focus on or disregard specific inputs (L066)", or "We specifically engineer a new outlier filter based on Mamba that can more effectively filter out mismatches by discarding particular inputs (L095)".

These propositions would be acceptable, if not for their contrast to how attention is described - "this soft attention is allocated across all data throughout, leading to attention leakage and the retention of redundant context (L057)", "..learning an attention map...suffer from attention leakage... (L123)". Assuming that the authors are using the term 'attention leakage' to address situations where false correspondence still receive 'at least a little' attention, this inherently happens in Mamba networks as well. The term 'selectivity' in Mamba networks refer to the idea of having its parameters ($\overline{B}, \overline{C}, \Delta$) dependent on the input - and for less important inputs, it would ideally mean that the step size $\Delta$ is smaller.  While this facilitates effective compression of context, 'smaller' is not really 'zero' - and I think this 'small' can also be regarded as 'attention leakage'.

Therefore, to the questions considered in the paper (L061) :: (i) attention mechanisms also selectively treat inliers and outliers, and allocate attention appropriately to different points, as effective as Mamba mechanisms do. (ii) Attention mechanisms also 'fully capture the information'. But yes, context compression happens in Mamba networks but not in vanilla attention networks. The manuscript should not overstate what Mamba networks can do, and not understate what attention networks can do.

**3. Misleading explanation regarding the efficiency and functioning of Mamba.**

L68: "...and employs hardware scanning methods. This presents a potential avenue for addressing (ii)", where (ii) is regarding context compression. Hardware-aware algorithms in Mamba address the efficiency, not the compression of context.

L175, equation 3 explanation: While it is true that the discretized SSM can be represented as convolution, that is NOT the case for Mamba. With the integration of selectivity, the parameters of Mamba networks become input-dependent, and they cannot be represented as convolution (static fixed kernels) anymore. Therefore, the statement "This characteristic endows Mamba with the inherent capability for parallel computation" is definitely flawed. The authors make the same error in section 4.1, where they mention "This enables the network to inherit Mamba’s characteristics, allowing it to focus on the inputs from inliers while disregarding outlier information" (i.e., input dependence) and proceeds to represent the network through a convolution operation (i.e., input independence). This is not only misleading, but this makes me suspect that the authors do not have a solid understanding of the Mamba networks.

## Paper is overall hard to follow, with vague propositions at times

L25: "...our approach outperforms existing state-of-the-art methods on several visual tasks..." - the proposed method is for two-view correspondence, and this statement is too vague / overstating the achievements of MambaMatch.

It is mentioned throughout the paper that Mamba networks "lack channel awareness (L80)", "falls short in channel information acquisition (L150)", and "underperforms in cross-channel information access(257)". Apart from these three sentences sounding ambiguous when put together, the paper lacks context about **why this is so**. They just include a reference (Guo et al., 2024) - and when reading the referenced paper, it is mentioned that "channel attention is used to reduce channel redundancy caused by the excessive hidden
state number", which is validated by a Channel Activation Visualization (Figure 3(b) of Guo et al., 2024). The authors should at least provide more context, or provide similar visualizations. In particular, since the ablation results in Table 6 shows that simply applying C.A without Mamba results in worse results, at least 3 channel activation visualizations of (a) with LCEM only (b) with LCEM and Mamba only, and (c) with LCEM and CAMF should be provided to sufficiently evidence this claim.

The method section is not in order, and do not fully explain the overall pipeline of the paper. From Figure 2, it seems that the initial correspondence set is passed through Position-Context Initialization, then the MambaMatch layers which contain modules in the order of Local Context Enhancement, Channel-Aware Mamba Filter, and Cluster Sequence Block. These are just mentioned in the overview, but in sections 4.1 and 4.2, only the Channel-Aware Mamba Filter and Local Context Enhancement Modules are discussed. This makes the method very difficult to follow.

Also, the inlier predictor seems like it should be placed outside of the MambaMatch layers which is repeated 4 times, since the inlier predictor outputs tensors with channel 1 (1-D), while MambaMatch layers accept tensors with channel dimensions of 128 (128-D).

"...while the second pipeline is discussed appropriately in the analysis. (L133)" should guide the readers specifically to Table 4, where it shows compatibility with Matchers. Otherwise, it is hard to follow what the authors are referring to.

$\mathcal{M}$ in Equations 10 and 11 should all be differentiated, unless the same MLP is being used for all $\mathcal{M}$.

## Less major issues, including errors in writing.

- Lack of visualization of "MambaMatch focusing on or discarding particular inputs" along the 4 layers of MambaMatch layers. The qualitative results in the paper are mainly visualization of the matches after pruning.
- How is the pruning done? How are erroneous correspondences filtered out? Is there any sort of thresholding involved?
- Regarding Table 6, are all settings in the Table trained following the same settings as MambaMatch? This is just a wary question, I find it weird that the performance increase is so low for (ii) and (iii). Or were they just carried out from weights of (iv)?
- In Table 6, there should be a comparison when even LCEM is not used.


### Writing mistakes:
- 'Correspondence' capitalized in L107 and L126
- ZOH is explained twice in L162 and L169, but it is actually only carried out in the latter.
- Typo in caption of Figure 3: Channel -> Channel
- Skin to -> Akin to (L417)

**Questions:**

Some questions I have regarding the paper -

1. In equation 16, what does $\mathcal{L}$ refer to?
2. What does it mean that the key points are normalized “with image size [OR] camera intrinsic”?
3. How were the FLOPs measured? Many open-source FLOPs-measuring libraries fail to capture the metrics correctly, especially in the presence of hardware optimizations in xformers or mamba. Can you provide per-module FLOPs calculation of the given method?
4. What are the GPU vRAM usage across the methods being compared?
5. It seems that when using Matcher such as SP+SG, the performance is much higher compared to using SIFT+NN. How do they differ in terms of efficiency?

While I believe the idea of using Mamba networks for the task of two-view correspondence is interesting, and the performances are satisfactory, I find the weaknesses stated above strongly outweigh the strengths of the paper.

---

### Official Review · Reviewer_vJjC · 2024-11-04

**Soundness:** 2
**Presentation:** 3
**Contribution:** 3
**Rating:** 6
**Confidence:** 4

**Summary:**

This paper proposes MambaMatch, an outlier removal approach of image matching. MambaMatch is constructed with the prevalent Mamba layer to capture contextual information in an efficient manner. Specifically, Channel-Aware Mamba Filter (CAMF) is designed by combining the Mamba layer with a channel attention structure. Moreover, Local Context Enhancement (LCEM) is introduced to improve the ability to capture local context. Experimental results demonstrate the superiority of MambaMatch when combined with the SIFT keypoints.

**Strengths:**

1. Mamba is a potential architecture to achieve a good balance between the accuracy and efficiency of outlier removal. This paper provides a reasonable implementation and initially validates its effectiveness.
2. The organization and presentation of this paper are clear.

**Weaknesses:**

**1. The explanation of the first motivation is confusing.**

In Line 57, the authors discuss the problem of the existing attention-based methods with the statement: “However, this soft attention is allocated across all data throughout, leading to attention leakage and the retention of redundant context”. Then, in Line 87, the superiority of the proposed Mamba is introduced as: “our Mamba-based approach not only focuses on potentially correct matches with higher weights but also discards false matches, avoiding attention leakage and achieving context compression.

However, the selection mechanism of Mamba cannot guarantee to avoid attention leakage according to the equations (2), (3), (4), (5). The “attention weights” of Mamba are still soft rather than hard. It is confusing why Mamba can avoid the attention leakage problem appearing in the existing attention-based methods like Transformer.

**2. The comparison experiment is not sufficient.**

Experimental results in Table 1 demonstrate the superiority of MambaMatch when combined with the SIFT keypoints. Meanwhile, the improvement provided by MambaMatch is smaller when it is combined with some advanced Matchers as shown in Table 4. Therefore, it is unclear whether MambaMatch can outperform the existing outlier removal approaches when combined with the advanced Matchers.

**3. Some expressions are unclear.**

3.1. In Figure 1, the authors state that “The transition of the yellow hue from closer to red to closer to green signifies a progression from lower to higher weights.” However, the false matches fully disappear in Figure 1 (d), without the color closer to red or closer to green. What are the scores provided by “Mamba-based Filters” for the false matches?

3.2. In Line 212, the authors state that “Drawing on the works of Zhang et al. (2019) and Liu & Yang (2023), we combine the Cluster-Sequence Block in our framework without delving into intricate specifics.” The role or function of the Cluster-Sequence Block in MambaMatch should be briefly introduced, even if its technical detail is unnecessary.

**Questions:**

Please provide more discussions and experimental results to address the above weaknesses.

---

### Note · Authors · 2024-11-14

I have read and agree with the venue's withdrawal policy on behalf of myself and my co-authors.